# Modeling Adsorption and Optical Properties for the Design of CO_2_ Photocatalytic Metal-Organic Frameworks

**DOI:** 10.3390/molecules26103060

**Published:** 2021-05-20

**Authors:** Priscila Chacón, Joseelyne G. Hernández-Lima, Adán Bazán-Jiménez, Marco A. García-Revilla

**Affiliations:** Chemistry Department, Natural and Exact Sciences Division, University of Guanajuato, Noria Alta S/N, Guanajuato 36050, Mexico; p.chaconmartinez@ugto.mx (P.C.); joseelyne.hernandez@ugto.mx (J.G.H.-L.); jabazan@ugto.mx (A.B.-J.)

**Keywords:** MOFs, photocatalysis, CO_2_ reduction, environmental-remediation, bonding, QTAIM

## Abstract

Four Metal-Organic Frameworks (MOFs) were modeled (IRMOF-C-BF_2_, IRMOF-C-(2)-BF_2_, IRMOF-C’-BF_2_, and IRMOF-C-CH_2_BF_2_) based on IRMOF-1. A series of linkers, based on Frustrated Lewis Pairs and coumarin moieties, were attached to IRMOF-1 to obtain MOFs with photocatalytic properties. Four different linkers were used: (a) a BF_2_ attached to a coumarin moiety at position 3, (b) two BF_2_ attached to a coumarin moiety in positions 3 and 7, (c) a BF_2_ attached in the coumarin moiety at position 7, and (d) a CH_2_BF_2_ attached at position 3. An analysis of the adsorption properties of H_2_, CO_2_, H_2_O and possible CO_2_ photocatalytic capabilities was performed by means of computational modeling using Density Functional Theory (DFT), Time-Dependent Density Functional (TD-DFT) methods, and periodic quantum chemical wave function approach. The results show that the proposed linkers are good enough to improve the CO_2_ adsorption, to hold better bulk properties, and obtain satisfactory optical properties in comparison with IRMOF-1 by itself.

## 1. Introduction

Metal-Organic Frameworks (MOFs), also known as porous coordination polymers, are crystalline materials with a bi- or tri-dimensional structure. They constitute the bridge between micro- and mesoporous materials. Among its properties, MOFs can be used for gas storage, energy conversion, chemical sensors, drug delivery and catalysis [1]. In particular, MOFs have recently been studied for their photocatalytic properties, primarily focused on H_2_ production from water [1,2,3,4,5,6], degradation of organic pollutants [2,7], and CO_2_ photo-reduction [2,8]. IRMOF-1 (MOF-5) is a crystalline and stable MOF, reported in 2005 by Li and co-workers [9]. Even at relatively high temperatures (300 °C), IRMOF-1 displays the desired thermo-stability for a material to be used for real-life applications. Because of its wide superficial area and fixed pore volume, IRMOF-1 was purposed for gas storage. These two properties make IRMOF-1 attractive as catalytic platform for the design of new MOFs for CO_2_ photo-reduction. IRMOF-1 is reported as support for an active photo-catalyst [8], as precursor of a catalyst [10,11,12] or as part of composites [13] or post modified structures [14] with photocatalytic activity. It is known that IRMOF-1 is capable of performing charge transfer by means of its organic linkers [2,15,16] through a process called Metal Ligand Charge Transfer (MLCT). Such MOF has proved its efficiency in phenol photo-degradation [3].

In other hand, coumarin has been widely used in the construction of optical devices because of its high optical efficiency. Photo sensors, molecular markers for both qualitative and quantitative studies, among others hold a coumarin moiety as the basic chromophore unit [17,18]. Coumarin and its derivatives display a remarkable behavior in photon capture in the UV-visible region [17]. Nevertheless, using coumarin as linker in MOFs to be the source of energy for a chemical transformation of adsorbates remained unexplored until now. Considering the potential advantages of using coumarin, we purpose it as starting material in the design of aggregate value linkers for the central cluster (Zn_4_O(CO_2_)_6_) of IRMOF-1-based-MOFs.

A further modification to obtain a catalytic MOF, addressed in the present contribution, is based on the recent finding that Frustrated Lewis Pairs (FLPs) activate the hydrogenation of CO_2_ for the production of hydrogen-rich fuels [19,20,21,22]. FLPs are formed by a Lewis acid-base pair hindered to form an adduct, either sterically or geometrically. However, voluminous substituents—commonly used to avoid adduct formation— tend to increase the activation energy required for the FLPs catalysis [19]. To overcome this problem, Ye Jingyun and Karl Johnson proposed in 2015 to stabilize FLPs by anchoring them to MOFs in the linker’s structure, thus, using the macromolecule as a catalytic platform [23]. Among the advantages of FLPs-functionalized-MOFs are the potentially efficient recuperation of the catalyst, potentially good capture and conversion of CO_2_ processes using the same material, thermal stability of the catalyst and low activation barriers. Such characteristics suggest that CO_2_ photo-reduction is possible under mild conditions.

In the present contribution, we address the design of a multipurpose linker. Such linker must have a combination of cooperative effects to obtain the desired CO_2_ catalytic properties: efficient UV-VIS absorption properties, capability of CO_2_ hydrogenation and subsequent reduction, and the expected structural spacing for obtaining a stable MOF. Our hypothesis is that hybrid organic coumarin-FLP linker (C-BF_2_) connected with the central metallic cluster displayed in the IRMOF-1 will achieve CO_2_ photo-reduction in the presence of H_2_ and H_2_O. With these considerations, we aim to improve the photocatalytic activity of IRMOF-1 central cluster by reducing the energy required for photon absorption and providing a catalytic-active site for the guest molecules in the MOF. This work considers the boron atom of the BF_2_ substituent in the coumarin moiety as the acidic counterpart of the FLP, whereas the basic counterpart is either the carbonyl oxygen of the coumarin moiety or oxygen of the closest OCO bridge in the central cluster of the IRMOF-1 derivative. In both cases, the FLP is restricted by coumarin’s rigid ring and MOF rigid and stable structure, without the need for steric hindrance. In this way, once CO_2_, H_2_, and H_2_O molecules are adsorbed, they could interact through the captured energy from the photon absorption and consequently, the CO_2_ could be photo catalytically transformed into a less contaminant molecules and hopefully in useful products.

## 2. Methods

IRMOF-C-BF_2_ was constructed using the central cluster of IRMOF-1 (Zn_4_O(CO_2_)_6_), under the assumption that both IRMOF-C-BF_2_ and IRMOF-1 crystallize in a cubic arrangement. For this purpose, 3-BF_2_-coumarin derivatives (C-BF_2_) were linked by their C8 to the C atom of the OCO bridges. Only three ligands were added in the direction of the three Cartesian axes (primitive cell, Figure 1). Both C5 and C8 of C-BF_2_ were connected to vicinal central clusters. Periodic replication of the primitive cell for unit cell construction used the BAND-ADF software [24,25]. The parameters of the cell were fixed as a = b = c = 12.941 Å and α = β = γ = 90°, as reported for IRMOF-1 [9] (unit cell, Figure 1). In addition, IRMOF-C’-BF_2_ was built from IRMOF-C-BF_2_, by changing the location of the BF_2_ group to the C7 of the coumarin moiety. In a similar way, it was assumed that this MOF crystallized on a cubic arrangement with the same cell parameters of IRMOF-C-BF_2_. IRMOF-C-CH_2_BF_2_ was built from IRMOF-C-BF_2_, by adding a methylene moiety in C3, between the coumarin linker and the BF_2_ group. 

Finite models of adsorbed CO_2_, H_2_, and H_2_O molecules were analyzed to characterize optical and catalytic properties. All molecular DFT calculations used the 2014 version of Amsterdam Density Functional (ADF) program [26,27]. A Double-zeta polarized basis (DZP) and a Generalized Gradient Approximation with a Perdew–Burke–Ernzerhof functional (GGA: PBE), for the Exchange and Correlation estimation were employed in all calculations (in exception of Zn atoms, for which Triple-zeta polarized basis (TZP) was used). Calculations that included IRMOF central cluster, where consideration of relativistic effects was needed, were done with a scalar approximation—ZORA scalar [27,28,29]. The Self-Consistence Field convergence criterion was fixed to10^−7^.

Calculations of IRMOF-C-BF_2_, IRMOF-C’-BF_2_, and IRMOF-C-CH_2_BF_2_ with guest molecules adsorbed considered the primitive cell. To warrant the proper filling of the valence orbitals, H atoms were added in the coumarin ring and CH_3_ groups to C atoms in OCO bridges (Figure 2). In addition, the MOF’s structure was frozen, making the guests move freely; this approach has proven to be effective by Jingyun in a prior report [23]. For the modeling of adsorption of more than one guest molecule, the atoms of the MOF structure were feezed, such guest molecules were added successively. As a comparison, similar calculations were performed for IRMOF-1 and the correspondent linker (C-BF_2_ for IRMOF-C-BF_2_, C’-BF_2_ for IRMOF-C’-BF_2_ and C-CH_2_BF_2_ for IRMOF-C-CH_2_BF_2_), to evaluate the effects of the change of linker and the bond of the coumarin derivative to the metal cluster, respectively. Excited states were obtained using TD-DFT Davidson’s method. Geometry optimizations were done for the brightest states —those with higher oscillator strength (*f*)—using symmetry restriction [30,31,32]. 

The modeling for the unit cell employed the BAND-ADF software 2014 [25], and the same theory level considered for the primitive cells. The PBE functional was used, to be consistent with the primitive cell’s calculations and because of the good agreement with experimental data reported for band properties’ calculations of IRMOF-1, in particular the band gap and Density of States (DOS) [33]. Geometry optimizations were performed by freezing the MOF structure. Band structure calculations considered an interpolation level of 4 and the effective mass. A K-space vector of (3-3-3) was employed for the Brillouin-zone integration in DOS calculations. In addition, 1801 equidistant energy points—energy grid—for adequate observation of the band’s population were used, as reported previously for IRMOF-1 [33]. 

QTAIM has been successfully and widely used to understand the properties of shared-shell and closed-shell bonding [34,35,36,37,38,39,40]. Relevant characteristics of a bond can be well evaluated by some criteria, (i) the presence of a bond critical point (BCP), (ii) values of electron density (ρ) and Laplacian (∇^2^ρ) at BCP, (iii) change of charge or electron population (ΔN), (iv) change of energy, (ΔE), and (v) delocalization index DI(X,Y) [41,42,43,44,45,46]. QTAIM analysis was performed at M06-2X/6-311 + +G(d,p) level of theory [47] of single points for finite models to study the intermolecular interactions between IRMOF’s and guest molecules adsorbed, using AIMAll program for properties calculation [48], Multifwn [49] and VMD [50] suites of program for a visual purpose.

## 3. Results and Discussion

### 3.1. IRMOF-C-BF2

#### 3.1.1. Primitive Cell of IRMOF-C-BF2

Previous studies [23] limit the modeling of the CO_2_ photo-reduction properties to the organic linker to model a new MOF, thus, ignoring the effects of the metallic cluster in its core. To evaluate if such approximation is reliable, we compared the optimized geometries of H_2_ and CO_2_ adsorption for the most affordable excited state of the primitive cell with the correspondent isolated coumarin derivative linker. Intermolecular distances were mostly inferior for the linker modeled in the absence of the metallic cluster (See Appendix A).

Distances between the two guests and C-BF_2_ linker imply that their absorption is performed on the linker structure in absence of the metallic cluster; the observed distance between the oxygen atom of CO_2_ and the B atom is 3.29 Å, H atoms of H_2_ and the oxygen atom of the carbonyl group are at 2.65 Å. These data lead us to confirm that H_2_ is susceptible to interact with the basic part of the FLP (O), whereas CO_2_ prefers the acidic counterpart (B). As mentioned above, the intermolecular distances are larger for the IRMOF-C-BF_2_ finite model which considers the metallic cluster. An O atom of the CO_2_ shows a distance of 4.971 Å with the closest hydrogen of the H_2_ molecule. Moreover, it is evident that H_2_ is too far from the MOF’s structure, and CO_2_ is adsorbed close to one of the OCO bridges instead of the nearest linker (the observed distances of each O atom to the C of the closest OCO bridge are 3.867 Å and 4.319 Å). Two hypotheses to explain this particular behavior are: (1) there is a competition between the ligands for the association with the guest molecules; and (2) this behavior is caused by the antagonism of the two electron-rich sites of the MOF, the carbonyl oxygen and the oxygen atoms of the OCO bridges for the absorption of the guest molecules. This situation is more evident for the case of CO_2_ absorption.

To explain the preference of the guests for the OCO bridges for absorption we have performed an analysis of the Quantum Chemical Topology of the adsorption process using the Quantum Theory of Atoms in Molecules to obtain a quantitative insight from the quantitative binding analysis. We used the absorption geometries reported in the Appendix A. As mentioned previously, some criteria help to evaluate bond characteristics. IRMOF-C-BF_2_ complex properties are shown in Table 1. All intermolecular interactions that present bond path and BCP with guest molecules adsorbed are shown in molecular graphs in Appendix A. There are several intermolecular interactions with guest molecules, most of them show values for Van der Waals interactions (ρ = ~10^−3^ au, ∇^2^ρ > 0 au) [45,51,52]. Delocalization index is a measure of polarity bond, pairs of atoms in closed-shell interactions show delocalization indices close to zero meanwhile in covalent bond tend to 1 [44]. Most of the interactions in these systems agree with weakly bound interactions (DI(X,Y) < 0.05) [53,54,55,56]. Moreover, each interaction represents a positive contribution (destabilization) for guest molecules (ΔE > 0) and some of them are involved in a loss of electronic population (ΔN(X) < 0) [54]. In the next lines, interactions are quantitatively analyzed.

Adsorbed CO_2_ establish interaction with MOF’s oxygens (69.23%), MOF’s carbons (15.4%), or even MOF’s hydrogens (15.4%), all bond paths and BCP are shown in molecular graphs Appendix A. CO_2_ preference for OCO bridges is backed up for BCP. Moreover, Laplacian isosurface shows an electron density concentration at OCO oxygens (Appendix A). That explains how CO_2_ carbon is attracted to these domains. In this domain, CO_2_ oxygens are close to other oxygens and interacting with them (Appendix A). Moreover, CO_2_ can interact with H_2_ and H_2_O, some of such interactions agree with hydrogen bond criteria [57,58,59,60] (i.e., 0.002–0.034 au for electron density and 0.024–0.139 au for Laplacian, 140–180°, ΔE(X) > 0, ΔN < 0, DI(X,Y) < 0.05). Then, CO_2_ maintains its ability to establish diverse interactions in this domain, especially with MOF’s oxygen. Additionally, guest intermolecular interactions have a stabilizing effect over CO_2_.

Regarding to the water adsorption, water interact with MOF’s oxygens (75%) and MOF’s hydrogens (25%), all bond paths and BCP are shown in molecular graphs (Appendix A). Hydrogen bonds were found in the presence of a water molecule (IRMOF-C-BF_2_--H_2_O and IRMOF-C-BF_2_-H_2_ CO_2_ H_2_O) and values of ρ and ∇^2^ρ (Table 1) are within the expected range (i.e., 0.002–0.034 au for electron density and 0.024–0.139 au for Laplacian). Angles were corroborated (140°–180°) and distance is consistent with a strong hydrogen bond = 2.5–3.0 Å [57,58,59,60]; however, one of them is a non-classical hydrogen bond (IRMOF-C-BF_2_--H_2_O).

Regarding adsorbed H_2_, there are interactions displayed by H_2_ with MOF’s carbon, MOF’s hydrogens, but not with MOF’s oxygens (Appendix A). Additionally, dihydrogen bonds that are present in IRMOF-C-BF_2_--H_2_ CO_2_ and IRMOF-C-BF_2_---H_2_ CO_2_ H_2_O agree with topological criteria (i.e., angle= 90–171°, distance < 2.4–2.6, ρ = 10^–2^~10^−3^, ∇^2^ρ > 0 ua) [55,56].

Guest intermolecular interactions were calculated using the AIMAll program. For each system, intermolecular interactions between H_2_, CO_2_, and H_2_O were detected (see BCP’s in Appendix A). Most of the intermolecular interactions between guest molecules agree with hydrogen bond criteria (Table 2) [57,58,59,60]. Additionally, water and carbon dioxide show an intermolecular BCP (O---C) (Appendix A), whose properties are consistent with Van der Waals interactions [51,52].

##### Optical Properties of Primitive Cell of IRMOF-C-BF2 during Adsorption Process

To contrast the optical properties of IRMOF-C-BF_2_, C-BF_2_, and IRMOF-1, a calculation of the most affordable excited states was performed, considering CO_2_ and H_2_ as guest molecules. The brightest transition for the coumarin derivative is 25 times more probable when it is isolated (C-BF_2_, Table 3) than when it is bonded at C8 to the metallic cluster (IRMOF-C-BF_2_, Table 3). The energy of the transition, on the other hand, is 1.48 times greater for C-BF_2_ (4.0 eV) than for IRMOF-C-BF_2_ (2.7 eV). The optical properties of coumarin are appreciably decreased when it is attached to a MOF, considering the oscillator strength (*f*) of vertical photo-excitations. Nevertheless, there is a probable improvement of the optical properties displayed by IRMOF-C-BF_2_ compared with IRMOF. IRMOF-C-BF_2_ shows a smaller energy of excitation, which was localized in the visible part of the electromagnetic spectra (Appendix A)), such behavior is desirable for photocatalytic devices. Comparing the observed transitions in IRMOF-1 and IRMOF-C-BF_2_, it is clear that the change in the ligand’s nature has positive effects both in the energy of the brightest transition and its probability (given by a higher *f*) (Appendix A), Table 3). IRMOF-C-BF_2_ shows a transition 14.7 times brighter than for IRMOF-1, whose respective energies are 2.7 eV and 4.2 eV. This implies a noticeable improvement of the optical properties to activate photocatalysis when a coumarin-like replaces the benzenoid linker in IRMOF-1. 

Table 4 and Figure 3 show the optimized geometries of the brightest excited states of the adsorption of H_2_, CO_2_, and H_2_O in IRMOF-C-BF_2_. Structure of Figure 3a) shows the adsorption of CO_2_ close to an OCO bridge (4.028 Å between the C of the bridge and the nearest O of CO_2_) and to one of the coumarin-derivative-linkers (with a distance of 5.007 Å between the B and an O of the guest). Regarding the adsorption of H_2_ (Figure 3b) and H_2_O (Figure 3c), the distances between the guests and IRMOF-C-BF_2_ are larger than 6.00 Å. On the other hand, Figure 3d–f displays geometry optimizations that imply more than one guest molecule. The closest proximity of adsorbed molecules to the MOF structure and among them are shown in Figure 3d) for adsorption of H_2_ followed by CO_2_ and Figure 3e) for adsorption of CO_2_ and then H_2_O. In the case of H_2_ and CO_2_ adsorption (Figure 3d), the oxygen atoms of CO_2_ are located at 4.148 Å and 4.123 Å of the carbon of an OCO bridge of MOF; H_2_ molecule is located at 2.899 Å of adsorbed CO_2_. In the case of CO_2_ and H_2_O as guest molecules (Figure 3e), CO_2_ is localized near a carbon atom of a OCO bridge, at 4.388 Å; the water molecule is localized at 3.608 Å of one oxygen of the same OCO bridge. The bond angles and interatomic distances of adsorbed molecules are not affected in any of the structures. Regarding the adsorption of H_2_ followed by CO_2_ and then H_2_O (Figure 3f), the places where H_2_, CO_2_, and H_2_O adsorb are similar to those observed for the two guest molecules. Thus, the results involving more than one guest molecule suggest a reaction between the guest molecules, mediated by the MOF structure.

Regarding the vertical excitations, the ones with higher *f* value are singlet-singlet transitions. The brightest ones are shown in Table 5. As showed, *f* values for IRMOF-C-BF_2_ with and without guest molecules are, despite their magnitudes, 10 times greater than the respective ones for IRMOF-1. Furthermore, the energy required for electron excitation is 150% smaller for IRMOF-C-BF_2_ than for IRMOF-1 (the optimized geometries in the absence of guest molecules are taken as reference).

The information related to the photocatalytic capability of the IRMOF-C-BF_2_ can be obtained from the characterization of the Potential Energy Surface (PES), such study is in progress. With this regard, a characterization of some candidate transition states and maximums found along the optimization process of the excited states, are presented in the Appendix A. A relevant observation is that at such maximums, the absorbed guest molecules display large distortions of their geometry.

##### Change in Order of the Guests’ Adsorption for Geometry Optimization in IRMOF-C-BF_2_

We performed the adsorption of the guest molecules adding CO_2_ and then H_2_ to IRMOF-C-BF_2_, to compare the optimized absorption geometries at the brilliant excited states. Appendix A shows the resultant geometry, for which a slight difference can be perceived when compared to Appendix A, in which the guest molecules are added in the opposite order. CO_2_ is near three different atoms in the MOF structure (Appendix A): its carbon atom is placed in the middle of two oxygen atoms of different kind, the first one located at 3.721 Å is the carbonyl oxygen of a linker; the second one is located at 3.157 Å and forms part of the OCO bridge located just next to the C-BF_2_ linker. Regarding the oxygen atom of CO_2_, it is placed farthest to the central cluster and it is also near the OCO bridge, at 4.826 Å. In contrast with the original adsorption order, H_2_ then CO_2_, in this experiment adsorbed H_2_ is located between the adsorbed CO_2_ molecule and the central cluster of the primitive cell. The distance among both guests is 3.161 Å. It is remarkable that when CO_2_ is added first, both guests are absorbed closer to the C-BF_2_ linker. Finally, the energies of both geometries are similar, –608.917 eV when H_2_ is added first and –608.753 eV when CO_2_ is added first.

Regarding the binding displayed for these systems, AIM properties of BCP were obtained and intermolecular interactions were analyzed (Table 1 and Table 6). BCP and bond paths are shown in the molecular graph (Appendix A). Change in order addition results in different behavior in guest localization and intermolecular interactions. In the case of IRMOF-C-BF_2_---H_2_ CO_2_, molecular hydrogen is adsorbed first; it prefers to interact with the aromatic framework of coumarin (it is consistent with its behavior in IRMOF-C-BF_2_---H_2_, see Appendix A, where it interacts with aromatic ring). Then CO_2_ is added, it is attracted to OCO framework (Appendix A), CO_2_ is near H_2_ and they form an interaction that agrees with hydrogen bond criteria (Table 2). In the case of IRMOF-C-BF_2_---CO_2_ H_2_, carbon dioxide is adsorbed first; it is attracted to the OCO region and carbonyl of coumarin (this agrees with IRMOF-C-BF_2_---CO_2_, see Appendix A, where CO_2_ interacts with carbonyl groups and OCO region). Then H_2_ is added, and it interacts with the aromatic ring of coumarin and forms an O—H interaction that agrees with hydrogen bond criteria (Table 7) with CO_2_ (Appendix A). The number of intermolecular interactions of guest and IRMOF-C-BF_2_ is larger when CO_2_ is added first (Table 6) because the dioxide interacts with OCO and carbonyl frameworks. This behavior is present in IRMOF-C-BF_2_---CO_2_ H_2_O and IRMOF-C’-BF_2_---H_2_ CO_2_. In IRMOF-C-BF_2_---CO_2_ H_2_, most of intermolecular interactions are consistent with Van der Waals interactions (ρ = ~10^−3^ au, ∇^2^ρ> 0 au and DI(X,Y) < 0.05) (Table 7) [51,52].

#### 3.1.2. Unit Cell of IRMOF-C-BF2

Unit cell geometries with H_2_ and CO_2_ as guest molecules were optimized and evaluated using BAND-ADF to confirm the changes in the properties of the primitive cell. We have special interest on the geometries adopted by the guests and the structure of the valence and conduction bands of the semiconductor cluster in the center of each node.

As can be seen in Figure 4, when the geometry optimization is performed considering the bulk structure of the unit cell, the guest molecules are located slightly more distant to the Lewis acid and base moieties of the FLP. When CO_2_ is the only guest molecule (a), the distances between the CO_2_ molecules and one of the linkers are: 5.692 Å from one oxygen atom of CO_2_ and the closest boron of the linker (BO distance), and 4.642 Å for the carbon-oxygen distance (CO) between the CO_2_ carbon and the carbonyl oxygen. The OCO bridge and the guest are located at 4.795 Å (CO distance between the carbon of CO_2_ and the closest oxygen of the indicated bridge). In addition, there is no change in bond angles or distances in CO_2_. When both H_2_ and CO_2_ are present (b), the distances between CO_2_ and the MOF structure are slightly increased: the CO distance between the carbon atom of CO_2_ and the closest oxygen atom in the OCO bridges is 5.722 Å, whereas the OB distance to the closest BF_2_ group is 6.568 Å. The guest molecules are located at 3.403 Å from each other, a value similar to the one obtained for calculations on the primitive cell. As in the case of the single guest adsorption, there are no significant changes in the bond distances and bond angles in the guest’s structures.

According to previous theoretical works, the band gap of IRMOF-1 is decreased when halogens are added to the linkers [61] or when the linkers are longer containing more conjugated carbon atoms [62]. Consequently, the band gap of IRMOF-C-BF_2_ is expected to decrease in a similar way, resulting in an improvement of the conduction properties of the MOF. The band properties of the isolated unit cell and the unit cell in the presence of guests are reported in Table 8. For the isolated unit cells, the band gap is considerably decreased from 3.4 eV to 0.403 eV (88.2 %); this is consistent with a previous report for linkers with more conjugated carbon atoms [62]. Such result implies a semiconductor behavior, with an increased facility for electronic conduction. The DOS of IRMOF-C-BF_2_ shows the contribution of the different atoms in both the valence and conduction bands (Appendix A). The major contribution to the valence band comes from the most electronegative atoms located in both the central MOF cluster and linkers (O atoms), whereas Zn and C atoms in the MOF cluster have both major contributions to energies surrounding the Fermi level (Appendix A)). This can be explained as a result of the influence of the electronegativity and the atomic effective charge on the shielding of the valence orbitals, the latter holds for the case of the Zn atom. The C atoms in the linker c) show a similar pattern to the C atoms in the cluster, with an appreciable contribution to both the valence and conduction bands, majorly due to their electronegativity and the amount of them in the MOF’s structure. The FLP proposed behaves according to the assumption that the B atoms would act as Lewis acids (with greater contribution to the conduction band) and the O atoms as Lewis bases (with greater contribution to the valence band) d). Considering the proximity of the energy levels generated by O atoms, it seems reasonable that the C atoms of CO_2_ associate to both the carbonyl and OCO bridge’s oxygen atoms. 

Comparing the energy parameters for the conduction and valence bands (Table 8), addition of a CO_2_ guest decreases the band gap (6.2% of decrease, from 0.403 eV to 0.378 eV) keeping the limits of both bands almost intact (- 4.789 to −4.408) eV. The major changes takes place when both guest molecules, H_2_ and CO_2_, are introduced to the MOF: an augment in the band gap (77.2% of augment, from 0.403 eV to 0.714 eV) resulting from a decrease in the bottom limit of the valence band (to −5.089 eV) and growth in the top of the conduction band (to −4.354 eV).

To analyze intermolecular interactions, QTAIM properties were calculated. However, to use less computing resources, only rigid fragments were evaluated. Molecular graphs show bond paths and BCP of intermolecular interactions in each system (Appendix A). Comparing bulks and isolated frameworks (IRMOF-C-BF_2_ unit cell CO_2_ with IRMOF-C-BF_2_---CO_2_ and IRMOF-C-BF_2_ unit cell CO_2_ H_2_ with IRMOF-C-BF_2_---H_2_ CO_2_), it was found that some interactions changed, then some BCP properties. One clear example of this is observed in O---O interaction with CO_2_ and carbonyl oxygen (see Table 1 and Table 9) where a decrease in ρ, ∇^2^ρ and DI(X,Y) is observed in bulk structure when is compared with the isolated framework, which means less bonding strength. This behavior is consistent with an increase in distance [44]. Additionally, in the presence of H_2_ and CO_2_, there are more intermolecular interactions in bulk system than in isolated framework (Table 1 and Table 9). These new intermolecular interactions come from H_2_ and bulk atoms, with OCO region, coumarin carbon (H---π), and carbonyl region. Moreover, one of its two H---O interactions (Table 9 and Appendix A) agrees with hydrogen bond criteria [57,58,59,60]. On the other hand, CO_2_ has two Van der Waals interactions [51,52].

Guest intermolecular interactions were obtained using the AIMAll program. Intermolecular interactions were detected between H_2_ and CO_2_ (Appendix A). Such interaction is formed between H---O; however, it does not agree with hydrogen bond criteria, mainly because of its angle (89.15°) (Table 10, it corresponds to a Van der Waals-like interaction [51,52]. This assumption is supported by topological properties. For example, if we compare the interactions with those for the isolated unit cell, Table 10 vs. Table 2, it is evident that binding properties of H---O such as electron density, Laplacian, and delocalization index are larger in magnitude.

### 3.2. IRMOF-C’-BF_2_

#### 3.2.1. Primitive Cells of IRMOF-C-(2)-BF_2_ and IRMOF-C’-BF_2_

Based on the previous observations that both H_2_ and CO_2_ adsorb preferentially close to the central cluster of IRMOF-C-BF_2_, a new linker was designed. Appendix A shows that CO_2_ adsorbs near one of the OCO bridges of IRMOF-C-BF_2_. With this regard, a BF_2_ group added on carbon C7 of the coumarin moieties will form a new FLP (formed by one oxygen atom of one OCO bridge and the boron on the coumarin moiety). Such FLP could be more suitable for guest adsorption over the pair formed by the carbonylic oxygen and the BF_2_ group C3 carbon of the linker.

To prove this hypothesis, a MOF structure with BF_2_ groups on carbons C3 and C7 is modeled (IRMOF-C-(2)-BF_2_). Geometry optimizations for H_2_ and CO_2_ adsorbed in IRMOF-C-(2)-BF_2_ were carried out. Figure 5b) shows the results of the competition for guest molecule’s absorption in the MOF structure. The BF_2_ group placed closer to the OCO bridges of the central cluster is preferred over the one in the α-position to the carbonyl groups. The OB distance between CO_2_ and the closest B (BF_2_ group) is 4.462 Å. The carbon of adsorbed CO_2_ is located at 3.793 Å of one of the oxygen atoms of the OCO bridge connected to the same linker as BF_2_, which the molecule is interacting with. This indicates that CO_2_ is being absorbed by the FLP built to include the OCO bridge. In addition, the guests are at 2.807 Å from each other (measuring the distance between the closest atoms of each guest, CO_2_ oxygen and H_2_ hydrogen); notice that the oxygen atom of CO_2_ that is located closest to H_2_ is not involved with the new FLP mentioned above. The hydrogen in H_2_ that is farthest of CO_2_ is observed almost in the middle of both oxygen atoms of the ester of another linker (3.363 Å from the O in the ring and 4.203 Å of the carbonylic oxygen). 

An important question to be solved is: is the adsorption near OCO bridges improved by the BF_2_ groups not directly involved in it? To give an answer to this question, the MOF structure IRMOF-C’-BF_2_ was built (Figure 5c), maintaining the BF_2_ groups on C7 of the coumarin moieties but eliminating the other ones. Geometry optimizations for the IRMOF-C’-BF_2_ were performed, maintaining the addition order of the guest molecules (H_2_ followed by CO_2_). Figure 5c shows the guests as close to each other as when adsorbed on IRMOF-C-BF_2_. The distances among them are 3.193 Å for the HO interaction (measured between H_2_ and the closest oxygen of CO_2_) and 3.364 Å for the HC interaction (measured between H_2_ and the carbon atom of CO_2_). These values are smaller than the ones observed for IRMOF-C-BF_2_ (a), 4.971 Å for the HO distance) and larger than those observed for IRMOF-C-(2)-BF_2_ (b), 2.807 Å for the HO distance). Nevertheless, H_2_ is adsorbed in such place in IRMOF-C’-BF_2_ not just because of its proximity to CO_2_, but in response to the presence of the carbonyl oxygen atom in the closest linker (displaying a HO distance of 2.709 Å). Moreover, the proximity of H_2_ to atoms other than those in CO_2_ is not as remarkable as for IRMOF-C-(2)-BF_2_. In addition, there is a distance of 4.241 Å between an oxygen atom of CO_2_ and the carbon atom of the closest OCO bridge. It is noticeable that when a BF_2_ group is placed near the OCO bridges (b,c)) the guest molecules are adsorbed in the middle of two linkers. 

Until this point, the proximity of the guests-MOF is comparable for both IRMOF-C-(2)-BF_2_ and IRMOF-C’-BF_2_. In addition, their absorption is apparently better (at closer proximity with the MOF structure) than in IRMOF-C-BF_2_. To discern the most suitable MOF for H_2_ and CO_2_ absorption, the excited states that could lead to electronic vertical excitations were obtained (Table 11). For the brightest transition, the probability estimated for the three MOF has the same magnitude order (10^−3^), although the probability calculated for IRMOF-C-BF_2_ is slightly greater than the one for IRMOF-C-(2)-BF_2_ and IRMOF-C’-BF_2_ (2.5 × 10^−3^ and 6.3 × 10^−3^, respectively). The energies of the optimized transitions show an interesting behavior (Table 11): for IRMOF-C-(2)-BF_2_, the energy of the brightest transition is decreased (to 1.96 eV, 27.4% of decrement). In contrast, the energy of such transition in IRMOF-C’-BF_2_ is augmented (to 3.0 eV, 11.1% of increment). This implies that the gap between the ground and the excited state is bigger when there is just one B atom per linker than when there is more than one. This is caused by an augment of the ground state’s energy in the case of two BF_2_ groups on the coumarin moiety. Furthermore, the electrons on the HOMO are more stabilized if the boron atoms are located closer to the central cluster of the MOF (Table 8). The energies of the bottom of the valence band are – 5.089 eV for IRMOF-C-BF_2_ (B on C3) and – 6.476 eV for IRMOF-C’-BF_2_ (B on C7). This observation could be explained by the increased contribution of fluorine atoms to the valence band when they are closer to the metallic cluster (Appendix A). If the valence and conduction bands are more separated in IRMOF-C’-BF_2_, it is evident that the energy of its brightest transition will be larger.

Intermolecular interactions of both systems (IRMOF-C’-BF_2_ and IRMOF-C-(2)-BF2) were analyzed; AIM properties of BCP were obtained (Table 12). BCP and bond paths are showed in molecular graphs (Appendix A). Energy contribution, E(X), points out to destabilization contribution from guest molecules, especially in CO_2_ molecule. The delocalization index agrees with noncovalent interactions (<0.05) [53,54], Laplacian, and electron density agree with Van der Waals interactions [51,52].

Absorbed CO_2_ shows interactions with MOF’s oxygens, MOF’s carbon, and even with H_2_ (Appendix A). BCP pointed out the interaction between CO_2_ and H_2_, in agreement with hydrogen bond criteria in IRMOF-C’-BF_2_ [57,58,59,60]. CO_2_ is attracted to the carbonyl domain and can interact with coumarin oxygens in IRMOF-C’-BF_2_, but this attraction is weaker in IRMOF-C-(2)-BF_2_. Additionally, parameters are similar to those observed in CO_2_ in a previous system (Table 1, IRMOF-C-BF_2_---CO_2_ H_2_O), but the number of interactions is fewer, because in IRMOF-C-BF_2_---CO_2_ H_2_O carbon dioxide access more easily to OCO and carbonyl frameworks than CO_2_ in IRMOF-C’-BF_2_ and IRMOF-C-(2)-BF_2_. Laplacian isosurface shows electron density concentration in OCO and coumarin oxygens, where guest molecules are attracted.

As CO_2_, molecular hydrogen is attracted to the OCO framework too, it shows interactions with MOF’s carbon and one hydrogen bond interaction with coumarin oxygen in IRMOF-C’-BF_2_. Such hydrogen bond fully agrees with the common hydrogen bond criteria (Table 12) [57,58,59,60]. Furthermore, there are H---π interactions between H_2_ and coumarin in both systems.

Guest intermolecular interactions were analyzed using AIM descriptors. Intermolecular interactions were detected between H_2_ and CO_2_, see BCP in Appendix A. There are O---H interactions; however, one of these interactions in IRMOF-C’-BF_2_ does not agree with hydrogen bond criteria [57,58,59,60] because of its angle (112.87°) (Table 13). 

To compare the differences in the optical properties of the purposed MOF (IRMOF-C’-BF_2_) and its isolated linker (C’-BF_2_), a calculation of the excited states in UV and Visible regions was performed, using optimized geometries (Appendix A). Once again, it is clear the isolated linker structure (C’-BF_2_) shows brighter transitions (Table 3). Comparing the linker’s transitions (C’-BF_2_ and C-BF_2_), their energies are 5.4 eV for C’-BF_2_ with an *f* value of 0.305, and 4.0 eV for C-BF_2_ with an *f* value of 0.210. Although the energy is increased when changing the BF_2_ moiety from C3 to C7, it is weighted by a transition 1.4 times brighter. The different MOF structures are similar in energy (2.7 eV for IRMOF-C-BF_2_ and 3.0 eV for IRMOF-C’-BF_2_) and similar in ***f*** values (8.4 × 10^−3^ for IRMOF-C-BF_2_ and 6.3 × 10^−3^ for IRMOF-C’-BF_2_).

#### 3.2.2. Unit Cells of IRMOF-C’-BF_2_

Regarding the bulk properties, the band gap of the unit cell of IRMOF-C’-BF_2_ is decreased with respect to IRMOF-1 [30,32] (Table 8), such behavior is similar to the displayed by IRMOF-C-BF_2_. The calculated value of 2.413 eV implies a decrease of 29.0% with respect to IRMOF-1, which is much smaller than the one obtained for IRMOF-C-BF_2_ (IRMOF-C-BF_2_ displays 88.1% of decrement and a band gap of 0.403 eV). The energy of the bottom of the valence band and the top of the conduction band are 6.476 and 4.082 eV, respectively. Although the structure of the linkers in both purposed MOFs is similar, the placement of the boron atom closer to the OCO bridges and far from the carbonyl moiety in the coumarin ring causes an increase in the band gap. 

The information of the contributions of the different atoms that compose the MOF in its valence and conduction bands are placed in the DOS (Appendix A). The major contribution to the valence band comes from the most electronegative atoms both in the central cluster and linkers (oxygen atoms and fluorine atoms), whereas zinc atoms in the cluster have important contributions to the states surrounding the Fermi level (Appendix A)); this can be understood as the influence of the electronegativity for non-metal atoms, and in the case of the zinc atom by the effect of the atomic effective charge on the shielding of the valence orbitals. The carbon atoms in the central cluster exhibit a contribution oriented to the conduction band (b), nevertheless, there is an appreciable contribution of such carbon atoms to the valence band. The carbon atoms in the linker (c) follow a different pattern, they have an appreciable contribution to both the valence and conduction bands, majorly due to their electronegativity and their location on the electron-rich coumarin rings. The FLP behaves with the boron atoms as Lewis acids; they have a greater contribution to the conduction band. In addition, the oxygen atoms of the OCO bridges play as Lewis bases; they have a greater contribution to the valence band (d). Considering the proximity of the energy levels generated by oxygen atoms, it seems reasonable that the C atoms of CO_2_ prefer the association with the OCO bridge’s oxygen atoms. In addition, oxygen atoms of CO_2_ prefer to interact with carbon atoms from the OCO bridges and with the boron atoms of BF_2_, both with major contributions to the conduction band. 

DOS of IRMOF-C-BF_2_ (Appendix A) and IRMOF-C’-BF_2_ (Appendix A), have more populated valence bands; this difference in population is bigger for IRMOF-C’-BF_2_. Fluorine atoms of the linkers and carbon atoms of the OCO bridges show an appreciable difference when present in the two MOFs. Fluorine atoms contribute to both valence and conduction bands over a wide range of energies in IRMOF-C-BF_2_ but show a localized role in a few states of remarkable population density in IRMOF-C’-BF_2_. This might be related to a redistribution of the fluorine electron density when these atoms are located closer to the central cluster. Carbon atoms of the OCO bridges display an active role on the conduction band of IRMOF-C’-BF_2_, this is not observed in IRMOF-C-BF_2_. It is possible that the proximity of the fluorine atoms (electron-rich entities) of the linkers increase the energy of frontier orbitals due to electronic repulsion.

### 3.3. IRMOF-C-CH_2_BF_2_

#### Primitive Cells of IRMOF-C-CH_2_BF_2_

IRMOF-C-CH_2_BF_2_ was purposed as a MOF whose organic linker could be more suitable to be synthesized than the displayed by IRMOF-C-BF_2_. The optimized geometry when H_2_ and CO_2_ are added is shown in Figure 6. The proximity of the guest molecules to the MOF is comparable to that obtained for a similar treatment with the MOF purposed herein (IRMOF-C-BF_2_, Appendix A; IRMOF-C’-BF_2_, Figure 5c)). The proximity of both guests is of 3.317 Å (OH closest distance); each guest adsorption is influenced by two groups on the MOF. In addition, the hydrogen atom of H_2_, farthest of the CO_2_ molecule, is adsorbed close to two OCO bridges; the OH distances among them are 4.381 Å and 4.791 Å. Moreover, CO_2_ is located near a FLP in one of the linkers, its oxygen atom that is not in proximity of H_2_ is at 4.493 Å of boron. In addition, the C of CO_2_ is located at 3.332 Å of the carbonyl oxygen of the coumarin moiety. 

BCP’s in IRMOF-C-CH_2_BF_2_ were calculated to analyze intermolecular interactions with guest molecules (CO_2_ and H_2_), AIM properties of BCP were obtained (Table 14) and in the following lines, interactions are analyzed. 

As was observed in a similar system (i.e., IRMOF-C’-BF_2_---H_2_ CO_2_), in IRMOF-C-CH_2_BF_2_ CO_2_ and H_2_ are attracted to OCO domain (Appendix A). Interaction properties (Table 14) agree with no covalent interactions [51,52]. Molecular graph (Appendix A) shows the BCP’s and bond paths for some intermolecular interactions. CO_2_ interacts with coumarin oxygen and coumarin carbon, but with H_2_. On the other hand, H_2_ has direct contact with oxygens in OCO and coumarin oxygen. This last interaction agrees with hydrogen bond criteria [57,58,59,60]. 

Additionally, an intermolecular interaction between H_2_ and CO_2_ was found, and its topological properties are shown in Table 15. There is an O---H interaction, it does not agree with hydrogen bond criteria because its angle (90.48°), but its properties are consistent with noncovalent interactions [51,52].

The excited states of IRMOF-C-CH_2_BF_2_ in the presence of H_2_ and CO_2_ as guest molecules were calculated (Table 3, Appendix A)). In comparison to the previous linkers, the brighter excited state of C-CH_2_BF_2_ has a greater f value (0.331); its energy of 4.0 eV is similar to that of C-BF_2_, which implies that the inclusion of an electron deficient substituent, regardless of its nature (BF_2_ or CH_2_BF_2_), leads to a brighter state at 4.0 eV. It is remarkable, however, that the addition of a CH_2_ group increases the probability of the mentioned transition. Whit this regard, similar characteristics of the brighter states of C-CH_2_BF_2_ and C’-BF_2_ are observed, although it is clear that C’-BF_2_ requires less energy for the mentioned transition. The IRMOF-C-CH_2_BF_2_ has an important improvement in energy, comparing the energy needed for the transition, it is just 0.59 times the value reported for IRMOF-1. Such situation is not possible with IRMOF-C-BF_2_ or IRMOF-C’-BF_2_ (Table 3). In addition, the probability to reach the brighter transition is 8.8 times greater than the reported value for IRMOF-1, belonging to the same magnitude to the MOF purposed in this paper. 

### 3.4. Further Topological Analysis of All Possible Intermolecular Hydrogen Bond

In the framework of QTAIM a decrement of the kinetic energy density (G) at BCP is related to a depletion of the electron density, because this situation implies less repulsion among electrons. Moreover, this behavior is observed at the hydrogen bond BCP when the hydrogen bond distance is increased, d(H---O) [62,63,64]. In addition, the positive curvature of ρ at the BCP, λ_3_^CP^, shows a good correlation with d(H---O), at variance with ρ and ∇^2^ρ displaying large data dispersion. With this regard, Espinosa and co-workers proposed a formula to calculate G at the bond critical point G^CP^ using λ_3_^CP^ as a variable in a linear fitting, G^CP^ = 15.3(1) λ_3_^cp^. [63] Then, the strength of a hydrogen bond is unambiguously characterized by G^CP^, the stronger the hydrogen bond the stronger repulsive among electron at BCP, increasing λ_3_^CP^ and therefore G^CP^. To clarify hydrogen bond interactions between MOFs and molecular guest, additional kinetic energy (G^CP^) and positive curvature (λ_3_^CP^ ) analyses were performed (Appendix A), all possible intermolecular hydrogen bonds were considered. Graphical plot of λ_3_^CP^ vs. d(H---O) exhibits a similar behavior reported by Espinosa [63,64], (see Appendix A). Furthermore, G vs. d(H---O), Figure 7, displays the expected behavior reported by Espinosa. In base of G^CP^ values the complete set of hydrogen bonds observed for guest molecules adsorbed in our modeled MOFs are consistent with a closed-shell interaction. Moreover, three strong hydrogen bonds are displayed related with water adsorption, which λ_3_ values (6.6, 4.9, and 12.2) highlight over the rest. 

## 4. Conclusions

We modeled four MOFs (IRMOF-C-BF_2_, IRMOF-C-(2)-BF_2_, IRMOF-C’-BF_2_ and IRMOF-C-CH_2_BF_2_) based on IRMOF-1 and a linker based on Frustrated Lewis Pairs and coumarin moieties to confer photocatalytic properties to the MOFs. The four different linkers used: (a) a BF_2_ attached to a coumarin moiety at position 3, (b) two BF_2_ attached in the coumarin moiety at positions 3 and 7 C-(2)-BF_2_, (c) one BF_2_ attached to the coumarin moiety at position 7, and (d) one CH_2_BF_2_ attached to the coumarin moiety at 3 position. We observe that the adsorption of H_2_, CO_2_, and H_2_O is possible and that it is probable that the systems will display CO_2_ photocatalytic properties.

In addition, we find that the hydrogen molecule is susceptible to interact with the basic part of the FLP (O), whereas carbon dioxide prefers the acidic counterpart (B). Moreover, for the MOFs primitive cells modeled, adsorbed H_2_ is far from the MOF’s structure and CO_2_ is adsorbed near one of the OCO bridges instead of the nearest linker. Moreover, BF_2_ attached to the coumarin reduces the vertical transition probabilities of such chromophore. Isolated Coumarin-BF_2_ moieties display larger optical properties than when being linkers in a MOF. Nevertheless, C-BF_2_-like linkers improve optical properties of IRMOF-1 by itself, thus an opportunity in this direction is evinced. The characterization of the binding properties by means of AIM shows that the adsorption of the guest molecules is the consequence of the formation of relevant binding interactions, as hydrogen bonds, Van der Waals forces, and electron localization and delocalization interactions.

Structure and adsorption properties of synthesized MOFs could be different to the expected ones, for instance, displaying lower adsorption efficiency to the calculated [65]. For this reason, an important question to be solved is: have this Coumarin-BF_2_ linkers produce MOF defects that negatively affect the stability, adsorption, and catalytic properties, beyond the predictive computational capability? Whit this regard, we have an encouraging clue: similar linkers display common MOFs structures as was reported by Hendon et al. [66]. In addition, the same study presents modeling results in agreement with experimental structures of synthesized MOFs. Moreover, the characterization of changes of electron density during guest (CO_2_, H_2,_ H_2_O) adsorption is relevant to the understanding of the interactions responsible for the stability of guest-MOF complexes. Whit this regard, G(CP) and λ_3_ results are used to classify the strength of the displayed hydrogen bonds of adsorbed molecules, finding 3 relevant hydrogen bond interactions related with water adsorption. Currently, we focused on the analysis of behavior of ρ(**r**) positive curvature (λ_3_) and G(GP) at the bond critical point versus the topological distance, [63,64,66] along the guest adsorption reaction path.

The relevance of our present work relies on the plausibility of the improvement of photocatalytic properties of IRMOF using Coumarin-BF_2_ linkers. Nevertheless, future computational and experimental work is needed to study the changes of the interactions along the adsorption process and to test the photocatalytic efficiency of such new MOFs.

## Figures and Tables

**Figure 1 molecules-26-03060-f001:**
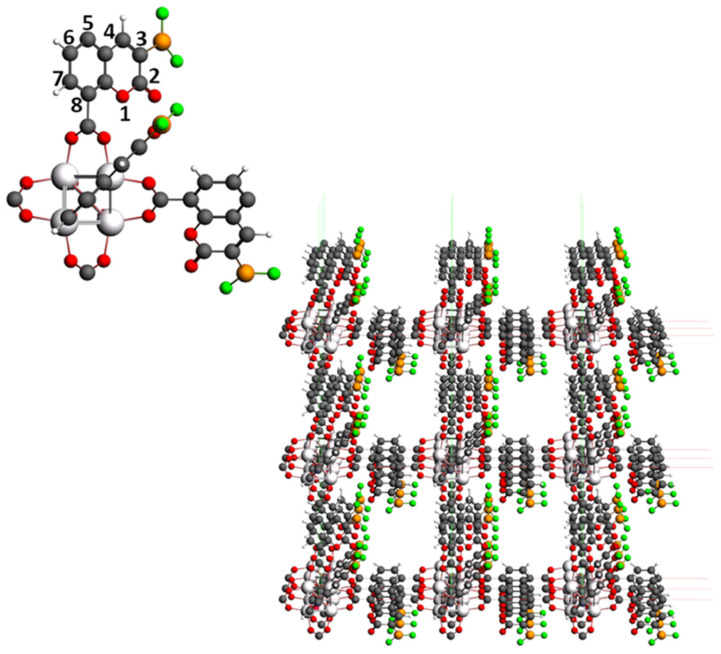
Unit cell of IRMOF-C-BF_2_ purposed in this work. The cell’s parameters considered are a = b = c = 12.941 Å and α = β = γ = 90°. At the left superior corner, primitive cell of IRMOF-C-BF_2_. Coordinates in xyz format are available in the Appendix A.

**Figure 2 molecules-26-03060-f002:**
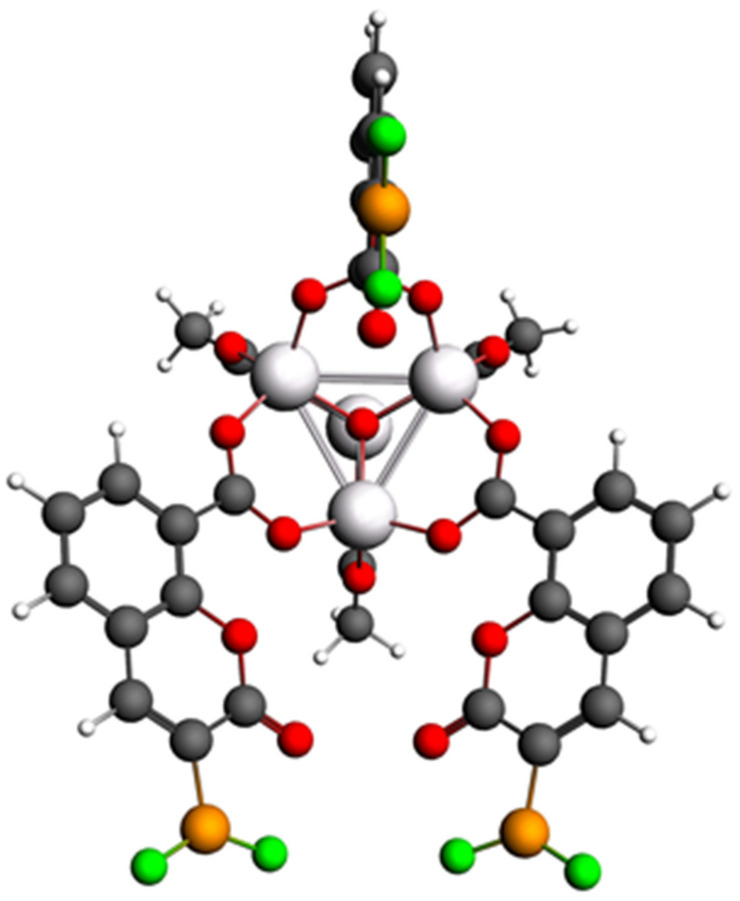
Structure of the primitive cell of IRMOF-C-BF_2_, after the proper filling with CH_3_ and H groups for OCO bridges and C5 position of the coumarin ring, respectively. Coordinates in xyz format available in the Appendix A).

**Figure 3 molecules-26-03060-f003:**
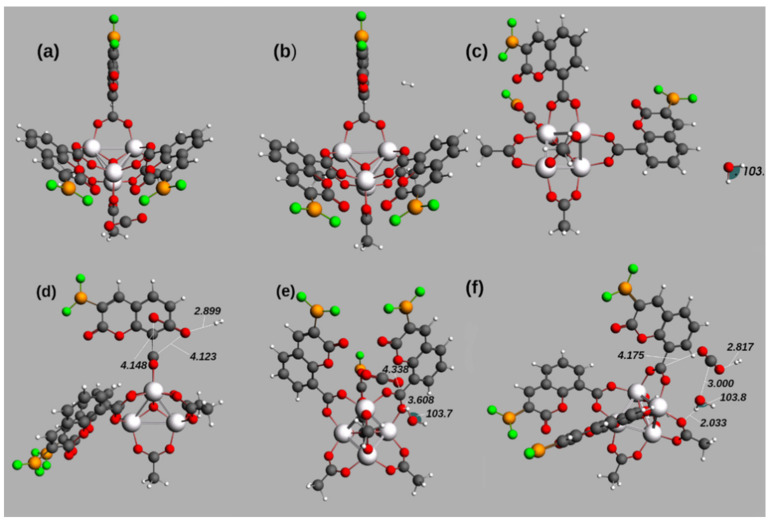
Optimized geometries of bright excited states reported in Table 5 for the interaction between IRMOF-C-BF_2_ and guest molecules. The energies for these are shown in Table 4. The structures presented are, considering the guest, as (**a**) IRMOF-C-BF_2_-CO_2_, (**b**) IRMOF-C-BF_2_-H_2_, (**c**) IRMOF-C-BF_2_-H_2_O, (**d**) IRMOF-C-BF_2_-H_2_-CO_2_, (**e**) IRMOF-C-BF_2_-CO_2_-H_2_O, (**f**) IRMOF-C-BF_2_-H_2_-CO_2_-H_2_O. The specified distances and angles are shown in Angstrom (Å) and degrees. Coordinates in xyz format available in the Appendix A.

**Figure 4 molecules-26-03060-f004:**
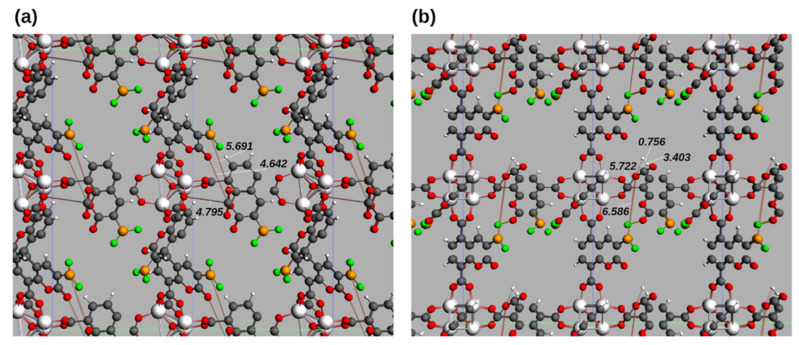
Minimum energy geometry for IRMOF-C-BF_2_ unit cell in the presence of (**a**) CO_2_ and (**b**) H_2_ and CO_2_ as guest molecules. The specified distances are shown in Angstrom (Å). Coordinates in xyz format available in the SI.

**Figure 5 molecules-26-03060-f005:**
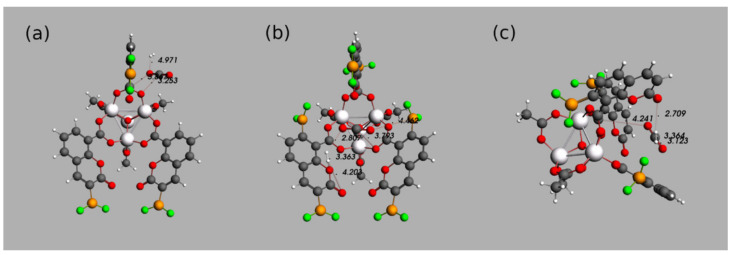
Comparison between the optimized geometries for the calculation of the excited states of the primitive cells (**a**) IRMOF-C-BF_2_, (**b**) IRMOF-C-(2)-BF_2_ y (**c**) IRMOF-C’-BF_2_, in the presence of H_2_ and CO_2_. The distances specified are shown in Angstroms (Å). Coordinates in xyz format available in the the Appendix A.

**Figure 6 molecules-26-03060-f006:**
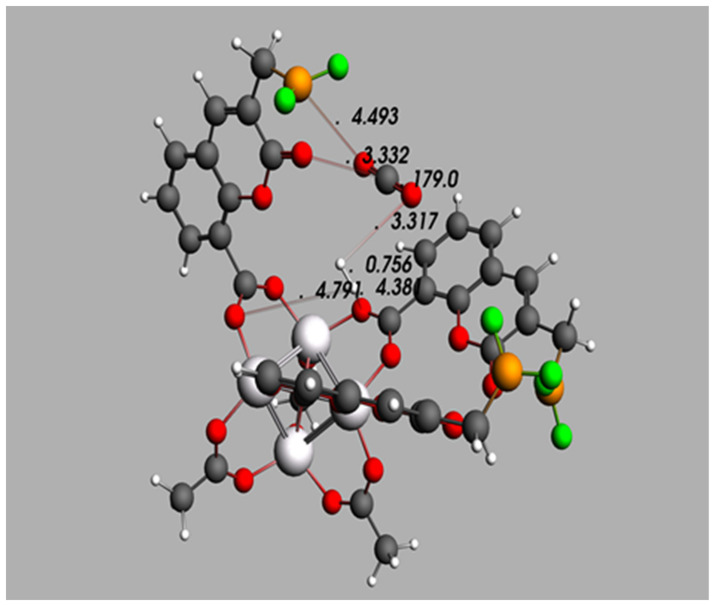
Optimized geometry for the absorption of H_2 H_ and CO_2_, in the primitive cell IRMOF-C-CH_2_BF_2_. The cited distances and bond angles are shown in Angstroms (Å) and degrees, respectively. Coordinates in xyz format available in the the Appendix A).

**Figure 7 molecules-26-03060-f007:**
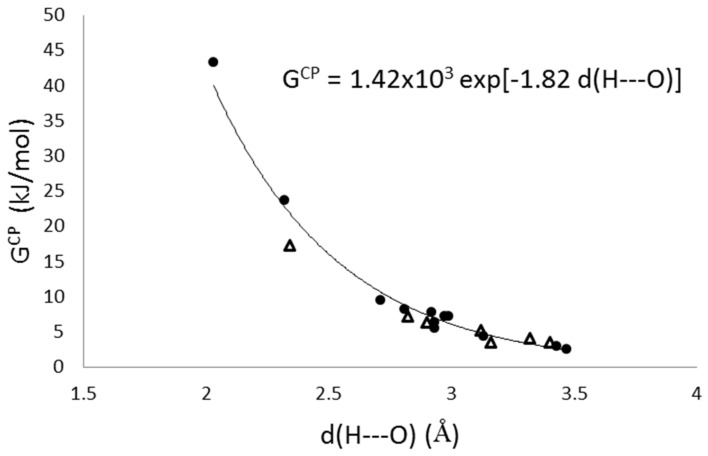
Behavior of G^CP^ versus d(H---O). Black dots are for hydrogen bonds between MOF´s and guest molecules. Triangles are for hydrogen bonds between guest molecules.

**Table 1 molecules-26-03060-t001:** Topological parameters of the BCP of bond in adsorption configurations of guest molecules (CO_2_, H_2_O and H_2_) on IRMOF-C-BF_2_.

Structure	Guest	Bond ^a^ X---Y	Distance (Å)	ρ	∇^2^ρ	DI(X,Y)	ΔE(X) kcal/mol	ΔN(X)
IRMOF-C-BF_2_-CO_2_	CO_2_	O---O	3.07	0.00771	0.02996	0.03868	91.27	−0.00446
		O---O	3.26	0.00381	0.01612	0.02143	92.83	−0.02477
		O---H	2.97	0.00379	0.01349	0.01334	92.83	−0.02477
		C---O	3.16	0.00528	0.02320	0.00999	37.41	0.03874
IRMOF-C-BF_2_-H_2_O	H_2_O	O---H	2.32	0.01168	0.04255	0.04561	71.77 ^b^	−0.00224 ^b^
IRMOF-C-BF_2_-H_2_	H_2_	H---C	3.03	0.00456	0.01150	0.01614	12.24	−0.00126
IRMOF-C-BF_2_-CO_2_ H_2_O	H_2_O	O---O	3.01	0.00823	0.03263	0.04285	69.81	0.02101
		O---O	3.22	0.00593	0.02241	0.03059	69.81	0.02101
	CO_2_	O---O	3.30	0.00348	0.01514	0.01894	91.68	−0.03609
		O---O	3.09	0.00599	0.02348	0.02581	89.88	0.01492
		O---O	3.55	0.00250	0.01147	0.00836	89.88	0.01492
		O---H	2.99	0.00373	0.01351	0.01197	91.68	−0.03609
		C---O	2.95	0.00798	0.03175	0.01660	43.12	0.03087
		C---O	3.02	0.00725	0.03000	0.01374	43.12	0.03087
IRMOF-C-BF_2_-H_2_ CO_2_	H_2_	H---H	2.70	0.00302	0.01053	0.00854	14.47	−0.01376
	CO_2_	C---C	3.26	0.00559	0.01996	0.00869	29.53	0.04988
		C---O	3.49	0.00264	0.01230	0.01181	29.53	0.04988
IRMOF-C-BF_2_-H_2_ CO_2_ H_2_O	H_2_	H---H	3.14	0.00105	0.00417	0.00314	14.95	−0.01398
	H_2_O	H---O	2.03	0.01879	0.07680	0.05326	16.71 ^b^	−0.03286 ^b^
	CO_2_	C---C	3.27	0.00539	0.01927	0.00828	36.07	0.04204

^a^ In bond column, X represents guest atom and Y MOF atom; ^b^ ΔE(X) and ΔN(X) in hydrogen bond are reported for H.

**Table 2 molecules-26-03060-t002:** Topological parameters of the BCP of bond in adsorption configurations of guest molecules (CO_2_, H_2_O and H_2_) on IRMOF-C-BF_2_. Guest intermolecular interactions.

Name	Guest	Angle	Distance (Å)	ρ	∇^2^ρ	DI(X,Y)	ΔE(X) kcal/mol	ΔN(X)	ΔE(Y) kcal/mol	ΔN(Y)
IRMOF-C-BF2-CO2 H2O	HH2O-OCO2	169.24	2.34	0.00848	0.03246	0.02483	16.09	−0.02908	89.88	0.01492
IRMOF-C-BF2-H2 CO2	HH2-OCO2	164.35	2.90	0.00324	0.01161	0.01484	14.47	−0.01376	92.89	−0.01609
IRMOF-C-BF2-H2 CO2 H2O	HH2-OCO2	174.57	2.82	0.00372	0.01317	0.01749	14.95	−0.01399	90.62	0.00061

**Table 3 molecules-26-03060-t003:** Energies and oscillator strengths of the brightest transitions of IRMOF-C-BF_2_ and C-BF_2_ in the presence of H_2_ and CO_2_ as probe guest molecules.

Structure Name	Energy/eV	*f*·10^−3^
IRMOF-1	4.2	0.57
IRMOF-C-BF2	5.4	8.4
C-BF2	4.0	210
IRMOF-C’-BF2	5.4	6.3
C’-BF2	3.0	305
IRMOF-C-CH2BF2	2.5	5.02
C-CH2BF2	4.0	331

**Table 4 molecules-26-03060-t004:** Primitive cell’s bonding energies of IRMOF-C-BF_2_ in the presence and absence of guest molecules.

Structure Name	Guest Molecule(s)	Relative Energy/eV
IRMOF-C-BF2	-	0.000
IRMOF-C-BF2-CO2	CO2	−22.912
IRMOF-C-BF2-H2	H2	−6.803
IRMOF-C-BF2-H2O	H2O	−14.096
IRMOF-C-BF2-CO2-H2	CO2 + H2	−29.661
IRMOF-C-BF2-CO2-H2O	CO2 + H2O	−37.117

**Table 5 molecules-26-03060-t005:** Optimized energies and oscillator strengths of the brightest transitions of IRMOF-C-BF_2_, in the presence and absence of guest molecules. A column including the results for IRMOF-1 is included as a comparison.

Guest Molecules	IRMOF-C-BF_2_		IRMOF-1	
	Energy/eV	*f*·10^3^	Energy/eV	*f*·10^−3^
-	2.7	8.3	4.1	0.64
CO2	2.7	5.6	4.1	0.42
H2	2.7	8.3	4.1	0.69
H2O	2.7	8.3	4.2	0.56
CO2 H2	2.7	8.4	4.2	0.57
CO2 H2O	2.5	4.3	4.2	0.43

**Table 6 molecules-26-03060-t006:** Topological parameters of the BCP of bond in adsorption configurations of guest molecules (CO_2_ and H_2_) on IRMOF-C-BF_2_---CO_2_ H_2_.

Name	Guest	Bond ^a^ X---Y	Distance (Å)	ρ	∇^2^ρ	DI(X,Y)	ΔE(X) kcal/mol	ΔN(X)
IRMOF-C-BF2---CO2 H2	H2	H--C	2.92	0.00624	0.01902	0.01642	7.96	0.01584
	CO2	C---O	3.16	0.00526	0.02315	0.00998	37.96	0.03723
		O---H	2.97	0.00378	0.01325	0.01322	1.34 b	−0.00624
		O---O	3.26	0.00379	0.01606	0.02123	92.68	−0.02581
		O---O	3.08	0.00768	0.02994	0.03838	91.29	−0.00058

^a^ In bond column X represents guest atom and Y MOF atom; ^b^ ΔE(X) and ΔN(X) in hydrogen bond are reported for H.

**Table 7 molecules-26-03060-t007:** Topological parameters of the BCP of bond in adsorption configurations of guest molecules (CO_2_ and H_2_) on IRMOF-C-BF_2_----CO_2_ H_2_. Guest intermolecular interactions.

Name	Guest	Angle	Distance (Å)	ρ	∇^2^ρ	DI(X,Y)	ΔE(X) kcal/mol	ΔN(X)	ΔE(Y) kcal/mol	ΔN(Y)
IRMOF-C-BF2---CO2 H2	HH2O-OCO2	157.24	3.16	0.00163	0.00708	0.00766	15.89	−0.01790	91.29	−0.00058

**Table 8 molecules-26-03060-t008:** Band properties for IRMOF-C-BF_2_ unit cell with and without the guest molecules H_2_ and CO_2_, and IRMOF-C’-BF_2_ without guest molecules. The values for IRMOF-1 are cited as reference [33,34].

Structure	Guest Molecules	Band Gap/eV	Valence Electrons	Valence Band Index	Conduction Band Index	Bottom of Valence Band/eV	Top of Conduction Band/eV
IRMOF-1	-	3.4	-	-	-	-	-
IRMOF-C-BF2	-	0.403	548	274	275	−4.789	−4.408
IRMOF-C-BF2	CO2	0.378	570	285	286	−4.789	−4.408
IRMOF-C-BF2	CO2, H2	0.714	572	286	-	−5.089	−4.354
IRMOF-C’-BF2	-	2.413	548	274	-	−6.476	−4.082

**Table 9 molecules-26-03060-t009:** Topological parameters of the BCP of bond in adsorption configurations of guest molecules (CO_2_ and H_2_) on fragments of unit cells.

Structure	Guest	Bond ^a^ X---Y	Distance (Å)	ρ	∇^2^ρ	DI(X,Y)	ΔE(X) kcal/mol	ΔN(X)
IRMOF-C-BF2 unit cell CO2	CO2	O---C	3.62	0.00282	0.01015	0.01167	98.13	−0.02345
		O---O	3.67	0.00118	0.00666	0.00844	101.69	−0.04197
		O---O	4.78	0.00014	0.00068	0.00149	98.13	−0.02345
IRMOF-C-BF2 unit cell CO2 H2	CO2	O---C	4.91	0.0001	0.00043	0.00033	76.99	−0.02534
		O---H	2.93	0.00298	0.01232	0.00889	76.99	−0.02534
	H2	H---O	2.93	0.00283	0.01044	0.01512	18.43 b	−0.02876 b
		H---O	3.43	0.00137	0.00639	0.00569	6.65 b	0.02471 b
		H---C	3.42	0.00177	0.00538	0.00534	6.65	0.02471

^a^ In bond column X represents guest atom and Y MOF atom; ^b^ ΔE(X) and ΔN(X) in hydrogen bond are reported for H...

**Table 10 molecules-26-03060-t010:** Topological parameters of the BCP of bond in adsorption configurations of guest molecules (CO_2_, H_2_O, and H_2_) on unit cell IRMOF-C-BF_2_. Guest intermolecular interactions.

Name	Guest	Angle	Distance (Å)	ρ	∇^2^ρ	DI(X,Y)	ΔE(X) kcal/mol	ΔN(X)	ΔE(Y) kcal/mol	ΔN(Y)
IRMOF-C-BF2 unit cell CO2 H2	HH2-OCO2	89.14	3.40	0.00171	0.00742	0.00738	18.43	−0.02876	76.99	−0.02534

**Table 11 molecules-26-03060-t011:** Energies and oscillator strengths of the brightest transitions of the optimized excited states for the absorption of CO_2_ and H_2_, in the cited order, into the primitive cells IRMOF-C-BF_2_, IRMOF-C-(2)-BF_2_ and IRMOF-C’-BF_2_.

Structure Name	Energy/eV	*f*·10^3^
IRMOF-C-BF2	2.7	8.4
IRMOF-C-(2)-BF2	1.96	2.5
IRMOF-C-BF2	3.0	6.3

**Table 12 molecules-26-03060-t012:** Topological parameters of the BCP of bond in adsorption configurations of guest molecules (CO_2_ and H_2_) on IRMOF-C-(2)-BF_2_ and IRMOF-C’-BF_2_.

Structure	Guest	Bond ^a^ X---Y	Distance (Å)	ρ	∇^2^ρ	DI(X,Y)	ΔE(X) kcal/mol	ΔN(X)
IRMOF-C-(2)-BF_2_	H_2_	H---C	3.07	0.00529	0.01608	0.01332	13.98	−0.01306
	CO_2_	O---C	3.36	0.00443	0.01573	0.01788	87.19	−0.01574
		O---C	3.18	0.00481	0.02022	0.01158	84.24	−0.00994
		C---O	3.01	0.00703	0.02939	0.01498	34.01	0.03829
IRMOF-C’-BF_2_	H_2_	H---O	2.71	0.00524	0.01680	0.02629	19.67 ^b^	−0.03312 ^b^
	CO_2_	H---C	3.07	0.00393	0.01011	0.01476	5.33	0.02927
		O---O	3.12	0.00610	0.02394	0.02616	86.48	−0.01103
		O---O	3.03	0.00700	0.02624	0.03210	87.84	−0.01642

^a^ In bond column X represents guest atom and Y MOF atom; ^b^ ΔE(X) and ΔN(X) in hydrogen bond are reported for H.

**Table 13 molecules-26-03060-t013:** Topological parameters of the BCP of bond in adsorption configurations of guest molecules (CO_2_ and H_2_) on IRMOF-C-(2)-BF_2_ and IRMOF-C’-BF_2_. Guest intermolecular interactions.

Name	Guest	Angle	Distance (Å)	ρ	∇^2^ρ	DI(X,Y)		ΔE(X) kcal/mol	ΔN(X)	ΔE(X) kcal/mol	ΔN(Y)
IRMOF-C-(2)-BF2	HH2-OCO2	141.45	2.81	0.00429	0.01474	0.02012		13.98	−0.01306	84.24	−0.00994
IRMOF-C’-BF2	HH2-OCO2	112.87	3.12	0.00270	0.01030	0.01307	5.33		0.02927	86.48	−0.01103

**Table 14 molecules-26-03060-t014:** Topological parameters of the BCP of bond in adsorption configurations of guest molecules (CO_2_ and H_2_) on IRMOF-C-CH_2_BF_2_.

Structure	Guest	Bond ^a^ X---Y	Distance (Å)	ρ	∇^2^ρ	DI(X,Y)	ΔE(X) kcal/mol	ΔN(X)
IRMOF-C-CH_2_BF_2_-H_2_ CO_2_	H_2_	H---O	3.13	0.00216	0.00855	0.01077	13.40 ^b^	−0.00557 ^b^
	H_2_	H---O	2.92	0.00432	0.01430	0.01754	11.98	−0.00067
	H_2_	H---O	3.47	0.00128	0.00549	0.00566	11.98	−0.00067
	CO_2_	O---O	3.19	0.00516	0.01941	0.02875	92.17	−0.01350
	CO_2_	O---C	3.43	0.00477	0.01459	0.01971	91.03	−0.01823

^a^ In bond column X represents guest atom and Y MOF atom; ^b^ ΔE(X) and ΔN(X) in hydrogen bond are reported for H.

**Table 15 molecules-26-03060-t015:** Topological parameters of the BCP of bond in adsorption configurations of guest molecules (CO_2_ and H_2_) on IRMOF-C-CH_2_BF_2_---H_2_ CO_2_. Guest intermolecular interactions.

Name	Guest	Angle	Distance (Å)	ρ	∇^2^ρ	DI(X,Y)	ΔE(X) kcal/mol	ΔN(X)	ΔE(Y) kcal/mol	ΔN(Y)
IRMOF-C-CH2 BF2 H2 CO2	HH2---OCO2	90.48	3.32	0.00204	0.00855	0.00873	13.40	−0.00557	91.03	−0.01823

## Data Availability

Not available.

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
