# Peer review of "Modeling Adsorption and Optical Properties for the Design of CO2 Photocatalytic Metal-Organic Frameworks"

_molecules, 2021, doi:10.3390/molecules26103060_

Round 1

Reviewer 1 Report

This article presents the modeling of four MOFs and the interaction with H2, CO2 and H2O. The article is well written and deserves publication once the following points are considered:

Point to consider:

Correct “M062X/ 6-311++g(d,p)” by “M06-2X/6-311++G(d,p)”

Have  the authors check a potential correlation between the interatomic distance and the electron density parameter in analogy to those described in the literature (see the work of Espinosa et al.)

Are the laplacian figures needed??

Correct “AIMA11” to “AIMAll”

Author Response

Guanajuato, México. May 3, 2020

Reviewer 1 

MDPI-AG-Molecules

We acknowledge the reviewer 1 for the observations, which helped to improve the quality of the manuscript. We have considered all the suggestions, a point-by-point answer is placed below:

Comments and Suggestions for Authors

This article presents the modeling of four MOFs and the interaction with H2, CO2 and H2O. The article is well written and deserves publication once the following points are considered: 

Point to consider:

Correct “M062X/ 6-311++g(d,p)” by “M06-2X/6-311++G(d,p)”

 ANSWER: We made the correction in the manuscript; the level of theory is properly written in the revised version of the manuscript.

Have the authors check a potential correlation between the interatomic distance and the electron density parameter in analogy to those described in the literature (see the work of Espinosa et al.)

 ANSWER: At the time of the first version of the manuscript we had not considered the correlation between the topological distance and electron density parameters. After the reviewer observation, we notice the nice behavior of the positive curvature (l3) of the electron density at the bond critical point in the characterization of changes of the electron density in hydrogen bonds. We think that the analysis of the l3 along the adsorption process in MOFs will give insight about the evolution of the electron density along the adsorption process, we just started to compute such changes using a series of geometries along the adsorption process. For this reason, we included the following comment in the conclusions with a reference to the Espinosa work.

“….. Besides, the characterization of changes of electron density during guest (CO2, H2, H2O) adsorption is relevant to the understanding of the interactions responsible of the stability of guest-MOF complexes. Currently, we are focused in the analysis of behavior of r(r) positive curvature (l3) at the bond critical point versus the topological distance, [66] along the guest adsorption.

Are the laplacian figures needed?

  ANSWER: We moved all QTAIM-figures to the supplementary information, without losing relevant information of topological descriptors, which is included in tables.

Correct “AIMA11” to “AIMAll”

ANSWER: We have corrected the typo in the entire manuscript.

We look forward to hearing from your decision on this contribution.

Sincerely yours,

Dr. Marco A. García-Revilla

Professor

Department of Chemistry

University of Guanajuato

36050-Guanajuato.

México.

Reviewer 2 Report

The manuscript deals with highly actual problem: design of new solid MOF-based materials for photocatalytic transformation. From this point the work is quite important. The authors used quantum-chemical calculation to model coumarine-contaning MOF. The geometry, electronic structure and possible interactions with CO2 and H2 are thoroughly modeled and discussed. The influence of the introduction of BF2 unit to organic linker on the potential properties of MOF are investigated. The influence of the coordination coumarine to material on the electronic structure of the system seems to be an interesting itself. The article can be published, but there are few notes:

1) It should be taken into the account, that real MOF always contans deffects, which are caused by the kinetics of the material formation. Of course, the introduction of coumarine will lead to the formation of deffects.  This fact, of course, is quite difficult to predict via calculation, but the deffects has significant (some times - determining) influence on the adsorbtion and catalytic properties of the real materials. From this point, the results obtained can be considered as preliminary: the authors showed the perspective structure which can be expected to provide desired photocatalytic rprocess when it will be obtained. This is quite important result, but it should be considered as part of more global investigation. The authors should add references to the examples of the prediction properties of MOFs by calculations which were followed by structure synthesis (if there are any).

2) The article seems to be little bit verbose, in my opinion it should be shorted to 18-20 pages. On the other hand, for the online edition it is not principal point.

Author Response

Guanajuato, México. May 3, 2020

Reviewer 2

MDPI-AG-Molecules

We acknowledge the reviewer 2 for the observations, which helped to improve the quality of the manuscript. We have considered all the suggestions, a point-by-point answer is placed below:

Comments and Suggestions for Authors

The manuscript deals with highly actual problem: design of new solid MOF-based materials for photocatalytic transformation. From this point the work is quite important. The authors used quantum-chemical calculation to model coumarine-contaning MOF. The geometry, electronic structure and possible interactions with CO2 and H2 are thoroughly modeled and discussed. The influence of the introduction of BF2 unit to organic linker on the potential properties of MOF are investigated. The influence of the coordination coumarine to material on the electronic structure of the system seems to be an interesting itself. The article can be published, but there are few notes:

  • It should be taken into the account, that real MOF always contans deffects, which are caused by the kinetics of the material formation. Of course, the introduction of coumarine will lead to the formation of deffects.  This fact, of course, is quite difficult to predict via calculation, but the deffects has significant (some times - determining) influence on the adsorbtion and catalytic properties of the real materials. From this point, the results obtained can be considered as preliminary: the authors showed the perspective structure which can be expected to provide desired photocatalytic rprocess when it will be obtained. This is quite important result, but it should be considered as part of more global investigation. The authors should add references to the examples of the prediction properties of MOFs by calculations which were followed by structure synthesis (if there are any).

ANSWER: We totally agree with the reviewer, for this reason we have included two references related with this observation and the following closing remark in the conclusions:

“Structure and adsorption properties of synthesized MOFs could be different to the expected ones, for instance, displaying lower adsorption efficiency to the expected. [64] For this reason, an important question to be solved is: have this Coumarin-BF2 linkers produce MOF defects that negatively affect the stability, adsorption, and catalytic properties, beyond the predictive computational capability? Whit this regard, we have an encouraging clue: similar linkers display common MOFs structures as was reported by Hendon et al [65]. In addition, the same study presents modeling results in agreement with experimental structures of synthesized MOFs. …”

“…The relevance our present work relies on the plausibility of the improvement of photocatalytic properties of IRMOF using Coumarin-BF2 linkers. Nevertheless, future experimental work is needed to test the photocatalytic efficiency of such new MOFs.”

2) The article seems to be little bit verbose, in my opinion it should be shorted to 18-20 pages. On the other hand, for the online edition it is not principal point.

ANSWER: We removed all QTAIM-figures to the supplementary information to reduce the length of the article (from 30 to 19 pages), the relevant information of topological descriptors is included in tables.

We look forward to hearing from your decision on this contribution.

Sincerely yours,

Dr. Marco A. García-Revilla

Professor

Department of Chemistry

University of Guanajuato

36050-Guanajuato.

México.
